# PROVABLE ADAPTATION ACROSS MULTIWAY DOMAINS VIA REPRESENTATION LEARNING

**Zhili Feng** *
Carnegie Mellon University
zhilif@andrew.cmu.edu

**Shaobo Han**
NEC Laboratories America, Inc.
shaobo@nec-labs.com

**Simon S. Du**
University of Washington
ssdu@cs.washington.edu

## ABSTRACT

This paper studies zero-shot domain adaptation where each domain is indexed on a multi-dimensional array, and we only have data from a small subset of domains. Our goal is to produce predictors that perform well on *unseen* domains. We propose a model which consists of a domain-invariant latent representation layer and a domain-specific linear prediction layer with a low-rank tensor structure. Theoretically, we present explicit sample complexity bounds to characterize the prediction error on unseen domains in terms of the number of domains with training data and the number of data per domain. In addition, we provide experiments on a two-way MNIST, a four-way fiber sensing dataset, and also the GTOS dataset to demonstrate the effectiveness of our proposed model.

## 1  INTRODUCTION

In many applications, collected datasets are cross-classified by a number of categorical factors such as a set of experimental or environmental conditions that are either controllable or known. These applications can be formalized as a *zero-shot domain adaptation* (ZSDA) problem with each domain being indexed on a multi-dimensional array. Domain-shifts often exist among data collected under different conditions, and to minimize the bias, one would like to collect diverse data to cover all conditions. However, due to high cost of data collection or insufficient resource in the field, it is often impractical to cover all possible combinations of factors. In most cases, training data collection can only cover a few limited, accessible scenarios, and the data from target domain may not be available during the training phase at all.

One motivating example is the distributed fiber optical sensing (Rogers, 1988) technique started to be used in smart city applications (Huang et al., 2019). Collecting fiber sensing data for machine learning is expensive and the characteristics of sensing data vary considerably across various heterogeneous factors, such as weather-ground conditions, signal source types, and sensing distances. Although conducting experiments to cover a small subset of combinations of different factors is possible, it is difficult to conduct experiments to cover all possible combinations. Specifically, in order to account for the effects of multiple factors (e.g., soil type × distance to sensor × weather), one has to visit multiple sites and set up the sensing equipment, and wait for the right time until the target type of data can be collected, and some combination of factors are not accessible due to the constraint of the physical world (e.g. a testbed at 10km sensing distance with asphalt pavement surface). Besides, in the digital pathology application, domain shift can be caused by different clinical centers, scanners, and tumor types (Fagerblom et al., 2021). It is impossible to cover every combination in training (e.g., one center maybe specialized in certain tumor class or own one particular type of scanners). Ideally, for these applications, one would want to train a model under a subset of scenarios, and adapt it to new *unseen scenarios* without collecting new data.

---

*Work done as an intern at NEC Labs America.

Another motivating application to design data augmentation with composition of operations. For instance, in image recognition applications, there are often multiple operations can be combined into augmented datasets, including blurring, translating, viewpoint, flipping, scaling, etc, and each with several levels or values to choose from. It may require huge amount of augmented data to cover all combinations of operations. Ideally, one only wants to try a small subset of all possible combinations. Previous works have shown that composing operations in careful ways outperforms randomly using data augmentation schemes (Ratner et al., 2017; Cubuk et al., 2019).

In the general ZSDA formulation, each domain is associated with a task descriptor, which is used to generate a predictor for the domain. This function that maps a task descriptor to a predictor, i.e., a predictor generator, is trained on training domains where data is available (Yang & Hospedales, 2014). In our setting, we can view the categorical indices for each domain as a task descriptor and thus apply existing approaches. However, this ignores the inherent multiway structure that connects different domains. Furthermore, existing approaches lack theoretical guarantees, and thus it is unclear under what conditions, the learned predictor generator can *provably* output an accurate predictor on unseen domains. A natural question is

> **Can we design a *provably sample-efficient* algorithm for zero-shot domain adaptation that exploits the multiway structure?**

## 1.1 OUR CONTRIBUTIONS

In this paper, we answer this question affirmatively. Our contributions are summarized below.

- we consider a multilinear structure which naturally characterize the relatedness among different domains. Formally, we assume that there are totally $D = d^M$ domains that form a $d^{\times M}$-dimensional task tensor, where the predictor on domain $t \in [D]$ is parameterized by $w_t^* \circ \phi^* \in \mathcal{W} \circ \Phi$ where $\phi^* : \mathcal{X} \to \mathbb{R}^p$ is a common representation for tasks and $w_t \in \mathbb{R}^p$ is a domain specific linear predictor. We only consider the case when $p$ is small. Note the linear predictors, $w_1^*, \ldots, w_D^* \in \mathbb{R}^p$, together form a form a $\underbrace{d \times d \times \cdots \times d}_{M \text{ times}} \times p$-dimensional tensor, which we denote as $\mathcal{T}$. To enable domain adaptation, we further impose a low-rank multi-linear on the linear predictor $\mathcal{T}$. The structure we impose on the model is that the slice along the last axis is always rank-$K$ with $K \ll d^M$. That is, for all $j \in [p]$, $\mathcal{T}_{\cdot,j} \in \mathbb{R}^{d \times d \times \cdots \times d}$ has rank $K$.

- We provide a finite-sample statistics analysis for this model. We show that if during training $T \in [D]$ source domains are observed, and in each observed domain $n$ samples are collected, then the expected excess risk across all $D$ domains is of order $\tilde{\mathcal{O}} \left( \left( \frac{TC(\mathcal{W}) + C(\Phi)}{nT} \right)^{1/4} + p \left( \frac{KdM^2}{T} \right)^{1/2} \right)$, where $C(\mathcal{F})$ represents the complexity of function class $\mathcal{F}$. The first term corresponds to the cost of learning the common representation and domain-specific linear classifiers, and the second term corresponds to the cost of generalizing to unseen domains. In the case where $n$ is large, we can simply ignore the first term, and conclude that $T$ should scale with $Kd(pM)^2$. This theoretically justifies our method can adapt to *unseen* domains with limited training domains.

- We test our proposed method on a two-way MNIST dataset and four-way fiber sensing datasets. The empirical results show our method matches the performance of general ZSDA algorithm by Yang & Hospedales (2014) which has a lot more parameters than our method.

## 2 RELATED WORK

For theoretical results of multi-task learning, Baxter (2000) first derived a covering number bound for sample complexities. Later Maurer et al. (2016) gave a uniform convergence bound for multi-task representation learning. Their bound is of order $\mathcal{O}(\sqrt{1/m} + \sqrt{1/T})$, where $m$ is the number of training samples in the target domain, so it is not reflecting how their learning process benefits from seeing multiple domains. Recently, Du et al. (2020); Tripuraneni et al. (2020) successfully derived results of type $\mathcal{O}(\sqrt{1/m} + \sqrt{1/(nT)})$, which satisfactorily explained how learning algorithm benefits from seeing abundant training data in many source domains, even if training data in target domain is scarce. This setting is often referred to as transfer learning or few-shot learning. A

key difference between this setting and ours is that we do not see any training data from the target domain. Blitzer et al. (2009) also studied ZSDA, but they studied a different approach which relies on multi-view learning and is restricted to linear models.

Many attempts to solving multi-task learning via low-rank matrices/tensors have been proposed Romera-Paredes et al. (2013); Signoretto et al. (2013); Wimalawarne et al. (2014), focusing on regularizing shallow models. Wimalawarne et al. (2014) specifically discussed task imputation and gave a sample complexity bound. However, their analysis assumed each training data is uniformly sampled from all source domains, so it is incomparable to our result. Yang & Hospedales (2017); Li et al. (2017) used low-rank tensor decomposition to share parameters between neural architectures for multiple tasks and empirically verified that both multi-task learning and domain generalization can benefit from the low-rank structure. Some different but closely related settings include unsupervised domain adaptation and zero-shot learning, where the test labels are either unseen during the training phase or different from what the model has been trained with. This is beyond the scope of this paper, as we only consider the case where all domains share same set of labels.

The low-rank matrix completion problem has been thoroughly studied in the past decade Srebro & Shraibman (2005); Srebro et al. (2004); Candès & Recht (2009); Recht (2011). Despite the combinatorial nature of rank, it has a nice convex relaxation, the trace norm, that can be optimized efficiently, in both noiseless and noisy settings Wainwright (2019). Although these quantities (trace norms, ranks, etc) are easy to calculate for matrices, most quantities associated with tensors are NP-hard to compute, including the trace norm Hillar & Lim (2013). There are many efforts in the theoretical computer science community to tackle tensor completion problems from the computational complexity perspective Barak & Moitra (2016); Liu & Moitra (2020). In this paper, we use the statistical properties of tensor completion despite their computational inefficiency. Hence our sample complexity bound does not contradict the seemingly worse bounds in the theory literature.

## 3 PRELIMINARY AND OVERVIEW

**Notation.** For $D \in \mathbb{Z}$, we let $[D] = \{1, 2, \ldots, D\}$. We write $\|\cdot\|$ to represent the $\ell_2$ norm or Frobenius norm. Let $\langle\cdot, \cdot\rangle$ be the inner product on an inner product space. Throughout the paper, we assume for simplicity in total there are $D = \underbrace{d \times d \times \cdots \times d}_{M}$ data domains. One can easily generalize the results to the general case where $D = \prod_{i=1}^{M} d_i$. We use $\otimes$ to denote the tensor product and $\odot$ the Hadamard product. We use $T \in [D]$ to denote the number of seen source domains, $n$ to be the amount of sample in a single source domain. Since each $t \in [D]$ also uniquely corresponds to a multi-index in $I \in [d]^{\times M}$, if $\mathcal{T} \in \mathbb{R}^{d \times \cdots \times d \times p}$ is a $d^{\times M}$ dimension tensor composing of $p$-dimensional linear functionals, we also write $\mathcal{T}_{t, \cdot} \in \mathbb{R}^p$ to represent the classifier at multi-index $I$, and we use $t$ and $I$ as indices interchangeably.

### 3.1 PROBLEM SETUP

During training, we have $T$ source domains out of the $D$ domains uniformly at random. For each domain $t \in [T]$, we have $n$ i.i.d data $\{(x_{t,i}, y_{t,i})_{i=1}^n\}$ samples from the following probability (Tripuraneni et al., 2020): $\mathbb{P}_t(x, y) = \mathbb{P}_{w_t^* \circ \phi^*}(x, y) = \mathbb{P}_x(x)\mathbb{P}_{y|x}(y|w_t^* \circ \phi^*(x))$, where all domains share the common feature mapping $\phi^* : (\mathbb{R}^r \to \mathbb{R}^p) \cap \Phi$ and a common marginal distribution $\mathbb{P}_x$. Each domain has its specific classifier $w_t^* \in (\mathbb{R}^p \to \mathbb{R}) \cap \mathcal{W}$. $\Phi$ can be a general function class and $\mathcal{W}$ composes of Lipschitz linear functionals. We denote this distribution as $\mathcal{D}_t$.

The key assumption is that $D$ domain-specific classifiers form a $D \times p$ rank-$K$ tensor $\mathcal{T}$ in the following sense:

$$\forall I \in [d]^{\times M} : \mathcal{T}_{I, \cdot}^* = w_t^* = \sum_{k=1}^K \overset{M}{\underset{m=1}{\odot}} \alpha_{k, t_m}^*, \tag{1}$$

for $\alpha_{k, t_m}^* \in \mathbb{R}^p$. We remark that this does not mean $\mathcal{T}$ has rank $K$. Instead, this means for each $j \in [p]$, $\mathcal{T}_{\cdot, j}$ has rank $K$.

Let $\ell : \mathbb{R} \times \mathbb{R} \to \mathbb{R}_{\geq 0}$ be the loss function, $L((w \circ \phi)(x), y) = \mathbb{E}_{x,y}[\ell((w \circ \phi)(x), y)]$ be the expected loss, $\widehat{L}_n((w \circ \phi)(x), y) = \frac{1}{n}\sum_{i=1}^n \ell((w \circ \phi)(x_i), y_i)$ be the empirical loss. When $x$ and $y$ are clear

from the context, we write $L(w \circ \phi)$ to denote $L((w \circ \phi)(x), y)$, and similarly for $\ell$ and $\widehat{L}_n$. When the number of samples $n$ is clear from the context, we write $\widehat{L} := \widehat{L}_n$. We write $L_t := \mathbb{E}_{\mathcal{D}_t}[\ell(\cdot, \cdot)]$ to emphasize the data is from $\mathcal{D}_t$. Our goal is to minimize the *average excess risk* across all domains:
$\frac{1}{D} \sum_{t \in [D]} \mathbb{E}_{\mathcal{D}_t} \left[ \ell \left( \hat{w}_t \circ \hat{\phi} \right) - \ell \left( w_t^* \circ \phi^* \right) \right]$.

Our learning algorithm first outputs the empirical risk minimizer (ERM) on the seen source domains:

$$\hat{w}_t, \hat{\phi} = \operatorname*{arg\,min}_{\substack{\phi \in \Phi, \\ w_1, \ldots, w_t \in \mathcal{W}}} \frac{1}{nT} \sum_{t=1}^{T} \sum_{i=1}^{n} \ell \left( (w_t \circ \phi)(x_{t,i}), y_{t,i} \right). \tag{2}$$

Naturally we put $\hat{w}_t$ into a $d^{\times M} \times p$ tensor $\tilde{\mathcal{T}}$, such that $\tilde{\mathcal{T}}_{t,\cdot} = \hat{w}_t$. Then we do a tensor completion for each $j \in [p]$ separately:

$$\widehat{\mathcal{T}}_{\cdot,i} = \operatorname*{arg\,min}_{\text{rank-}K \ \mathcal{T}} \frac{1}{T} \sum_{t \in [T]} \left| \mathcal{T}_{\cdot,i} - \tilde{\mathcal{T}}_{t,i} \right|. \tag{3}$$

Finally, at test time, for target domain indexed by $t \in [D]$ and test data $x \sim \mathcal{D}_t$, we make the prediction $\hat{y} = (\hat{w}_t \circ \hat{\phi})(x)$, where $\hat{w}_t = \widehat{\mathcal{T}}_{t,\cdot}$.

**Notion of Complexity**   We use Gaussian complexity and Rademacher complexity to measure the function class complexity.

**Definition 3.1** (Gaussian and Rademacher complexity). *Let $\mathcal{F} \subset \{f : \mathbb{R}^p \to \mathbb{R}^q\}$ be a vector-valued function class, the empirical Gaussian and Rademacher complexity on $n$ points $X = \{x_1, \ldots, x_n\}$ are defined as:*

$$\widehat{\mathfrak{G}}_n(\mathcal{F}_{|X}) = \mathbb{E}_g \left[ \sup_{f \in \mathcal{F}} \frac{1}{N} \sum_{i \in [N]} \sum_{k \in [q]} g_{ki} f_k(x_i) \right], \quad \widehat{\mathfrak{R}}_n(\mathcal{F}_{|X}) = \mathbb{E}_\epsilon \left[ \sup_{f \in \mathcal{F}} \frac{1}{N} \sum_{i \in [N]} \sum_{k \in [q]} \epsilon_{ki} f_k(x_i) \right],$$

*where $g_{ki}$'s are all standard Gaussian random variables, $\epsilon_{ki}$'s are Rademacher random variables, and $f_k$ is the $k$th entry of $f$.*

Let $\mathfrak{G}_n(\mathcal{F}) = \mathbb{E}_X[\widehat{\mathfrak{G}}_n(\mathcal{F}_{|X})]$ denote the expected Gaussian complexity (similarly for Rademacher complexity). To deal with the divergence among domains, we adopt the following notions from Tripuraneni et al. (2020): $\overline{\mathfrak{G}}_n(\mathcal{W}) = \max_{Z \in \mathcal{Z}} \hat{\mathfrak{G}}_n(\mathcal{W}_{|Z})$, where $\mathcal{Z} = \{(\phi(x_1), \cdots, \phi(x_n)) \mid \phi \in \Phi, x_i \in \mathcal{X} \text{ for all } i \in [n]\}$.

To measure the complexity of tensors, we use the notion of pseudo-dimension.

**Definition 3.2** (Pseudo-dimension). *Let $\mathcal{F} \subseteq \mathbb{R}^{\mathcal{X}}$ be a real-valued function class. Let $x_1^m = (x_1, \ldots, x_m) \in \mathcal{X}^m$. We say $x_1^m$ is pseudo-shattered by $\mathcal{F}$ is there exists $r = (r_1, \ldots, r_m) \in \mathbb{R}^m$ such that for all $b = (b_1, \ldots, b_m) \in \{\pm 1\}^m$, there exists $f \in \mathcal{F}$ such that $\text{sign}(f_b(x_i) - r_i) = b_i$ for all $i \in [m]$. The pseudo-dimension of $\mathcal{F}$ is defined as:*

$$\text{Pdim}(\mathcal{F}) = \max \left\{ m \in \mathbb{N} : \exists x_1^m s.t \ x_1^m \text{ is pseudo-shattered by } \mathcal{F} \right\}.$$

## 4   THEORETICAL RESULTS

In this section, we derive the sample complexity bound for our DG setting. As a high level overview, we first prove that on the $T$ seen source domains, with enough training data and sufficient regularity conditions, we can have $\hat{w}_t$ being sufficient close to $w_t^*$ on the source domains. Then we show that with a large enough $T$, even on an unseen domain $t \in [D]\backslash[T]$ we can also approximate $w_t^*$ well by completing the low-rank tensor formed by learned $\hat{w}_t$. Finally, by certain regularity conditions on the loss function, we can show the excess risk is also small.

We require the following regularity assumptions for our theoretical development.

**Assumption 4.1.**     *1. The learning problem is realizable, that is, $w_t^* \in \mathcal{W}$ for all $t \in [D]$ and $\phi^* \in \Phi$. WLOG assume that $w_t^* \circ \phi^*$ is the unique minimizer of $L_t$ for all $t$.*

2. *$\ell(\cdot, \cdot)$ is B-bounded, and $\ell(\cdot, y)$ is L-Lipschitz.*

3. *For all $w \in \mathcal{W}$, $\|w\| \leq W$.*

4. *$\sup_x \|\phi(x)\| \leq D_\mathcal{X}$, for any $\phi \in \Phi$.*

5. *For all $t \in [D]$, $L_t$ is $\lambda$-strongly convex in $w_t$ for $\phi^*$.*

The first assumption is indeed general. Since we care about the excess risk, in realizable problems we can compare to the unique minimizer of $L$ in $\mathcal{W} \circ \Phi$ instead of $w_t^* \circ \phi^*$. The existence of a unique minimizer is necessary, otherwise tensor completion can lead to arbitrary errors. Assumptions 2-4 are common. Lastly, without strong convexity in the last assumption, one cannot derive the closeness of $\hat{w}_t$ to $w_t^*$ from the closeness of the excess risk on source domains.

### 4.1 LEARNING COMMON REPRESENTATION

In this subsection, we discuss how to learn the shared representation. The proof follows Tripuraneni et al. (2020) with modifications to adopt to our setting. First we introduce some definitions that defines the divergence among the source domains and target domains.

**Definition 4.1.** *For a function class $\mathcal{W}$, T functions $w_1, \ldots, w_T$, and data $(x_t, y_t) \sim \mathcal{D}_t$ for any $t \in [T]$, the **task-average representation difference (TARD)** between representation $\phi$ and $\phi'$ is defined as:*

$$\bar{d}_{\mathcal{W},w}(\phi'; \phi) = \frac{1}{T} \sum_{t \in [T]} \inf_{w' \in \mathcal{W}} \mathop{\mathbb{E}}_{\mathcal{D}_t} \left( \ell(w' \circ \phi') - \ell(w_t \circ \phi) \right).$$

*Let $w^*, \phi^*$ be the true underlying functions. Define the **uniform absolute representation difference (UARD)** to be*

$$d_{\mathcal{W}}(\phi'; \phi) = \sup_{t \in [D]} \sup_{w_t \in \mathcal{W}} \left| \mathop{\mathbb{E}}_{\mathcal{D}_t} \left( \ell(w_t \circ \phi') - \ell(w_t \circ \phi) \right) \right|.$$

*For $\mathcal{W}$, we say $w = \{w_1, \ldots, w_T\}$ is $(\nu, \epsilon)$-**diverse** for a representation $\phi$, if the following holds for all $\phi' \in \Phi$:*

$$d_{\mathcal{W}}(\phi'; \phi) \leq \bar{d}_{\mathcal{W},w}(\phi'; \phi)/\nu + \epsilon.$$

The notion of task-average representation difference was introduced by Tripuraneni et al. (2020). In their setting, they bound the *worse-case representation difference (WCRD)* between two representations $\phi, \phi' \in \Phi$ which is defined as

$$\sup_{w \in \mathcal{W}} \inf_{w' \in \mathcal{W}} \mathop{\mathbb{E}}_{(x,y) \sim \mathbb{P}_{w \circ \phi}} [\ell(w' \circ \phi') - \ell(w \circ \phi)],$$

using the task-average representation difference. One should note that WCRD is changing the data distribution, while UARD takes expectation over the true distribution $\mathcal{D}_t$. We are saying that under the true distribution, the worst case difference between using $\phi$ vs. $\phi'$, over both the choice of linear classifiers $w$ and the domains, is not much larger than TARD. Intuitively, this says that our source domains and task domains are benign enough, such that the ERM $\hat{\phi}$ performs similarly to the true $\phi^*$.

Although it seems that $\bar{d}_{\mathcal{W},w}(\phi'; \phi)$ can be negative and $d_{\mathcal{W}}(\phi'; \phi)$ is always positive, later we only consider $\bar{d}_{\mathcal{W},w^*}(\phi'; \phi^*)$ and $w_t^* \circ \phi^*$ is assumed to be the minimizer. Hence, $\bar{d}_{\mathcal{W},w^*}(\phi'; \phi^*)$ is always positive, and our notion of task diversity makes sense.

We cite the following theorem from Tripuraneni et al. (2020).

**Theorem 4.1.** *Let $\hat{\phi}$ be the ERM in eq. (2). Under assumption 4.1, with probability at least $1 - \delta$, we have*

$$\bar{d}_{\mathcal{W},w^*}(\hat{\phi}, \phi^*) \leq 4096L \left[ \frac{WD_\mathcal{X}}{(nT)^2} + \log(nT) \cdot \left[ W \cdot \mathfrak{G}_{nT}(\Phi) + \overline{\mathfrak{G}}_n(\mathcal{W}) \right] \right] + 8B\sqrt{\frac{\log(2/\delta)}{nT}}.$$

## 4.2 Learning Domain Specific Linear Layer and Tensor Completion

Now we give results for learning the domain specific linear classification.

**Theorem 4.2.** *Let $\hat{w}, \hat{\phi}$ be the ERM in eq.* (2). *Let assumption 4.1 holds. With probability at least $1 - \delta$, we have*

$$\frac{1}{T}\sum_{t\in[T]}\|\hat{w}_t - w_t^*\|_2 = \mathcal{O}\left(\sqrt{\frac{2}{\lambda}}\left(L\left[\frac{WD_{\mathcal{X}}}{(nT)^2} + \log(nT)\cdot\left[W\cdot\mathfrak{G}_{nT}(\Phi) + \overline{\mathfrak{G}}_n(\mathcal{W})\right]\right] + 8B\sqrt{\frac{\log(2/\delta)}{nT}}\right)^{1/2}\right).$$

The proof of Theorem 4.2 starts with showing that $L(\hat{w}_t \circ \phi^*)$ is close to $L(w_t^* \circ \phi^*)$, while a normal ERM analysis only asserts $L(\hat{w}_t \circ \hat{\phi})$ is close to $L(w_t^* \circ \phi^*)$. Such assertion holds by the $(\nu, \epsilon)$-diverse assumption. Intuitively, such diversity assumption makes sure our learning landscape is somehow smooth: $\hat{w}_t$ should be stable such that if we perturb $\hat{\phi}$ to $\phi^*$, the loss function does not alter too much. Together with strong convexity, these conditions guarantee $\hat{w}_t$ is close to the true $w_t^*$ on average.

After we learn $\hat{w}_t$ for all $t \in [T]$, we find another ERM $\hat{\mathcal{T}}$ by constructing $p$ tensors $\hat{\mathcal{T}}_{\cdot,i}$ for all $i \in [p]$ separately as in eq. (3). We only observe the empirical tensor completion loss $\frac{1}{T}\sum_{t=1}^{T}|\hat{\mathcal{T}}_{t,i} - \tilde{\mathcal{T}}_{t,i}|$ during training. With large enough $T$, by a uniform convergence argument, we can show that $\frac{1}{T}\sum_{t=1}^{T}|\hat{\mathcal{T}}_{t,i} - \tilde{\mathcal{T}}_{t,i}|$ can approximate $\frac{1}{D}\sum_{t=1}^{D}|\hat{\mathcal{T}}_{t,i} - \mathcal{T}_{t,i}^*|$ fairly accurately.

**Lemma 4.1.** *Let $\mathcal{X}_K$ be the class of rank-$K$ tensor of shape $d^{\times M}$, its pseudo-dimension can be bounded by* $\text{Pdim}(\mathcal{X}_K) \le KdM^2\log(8ed)$.

$\text{Pdim}(\mathcal{X}_K)$ is computed by treating tensors as polynomials and counting the connected components of these polynomials. Even though any $X \in \mathcal{X}_K$ has $d^M$ entries, its complexity only scales as $\text{poly}(K, d, M)$. This assures that we only need to see polynomially many source tasks to perform well in unseen domains. Using the pseudo-dimension, we have the following uniform convergence result.

**Theorem 4.3.** *With probability at least $1 - \delta$,*

$$\frac{1}{D}\sum_{t\in[D]}\left\|\hat{\mathcal{T}}_{t,\cdot} - \mathcal{T}_{t,\cdot}^*\right\| \le \frac{1}{T}\sum_{t\in[T]}\sum_{j\in[p]}\left|\hat{\mathcal{T}}_{t,j} - \tilde{\mathcal{T}}_{t,j}\right| + p\sqrt{\frac{KdM^2\log(8ed) + \log(p/\delta)}{T}} + \tilde{\mathcal{O}}(n^{-1/4}).$$

$$(4)$$

The last $\mathcal{O}(n^{-1/4})$ term in eq. (4) comes from Theorem 4.2. This is the cost we pay for not knowing the true $\mathcal{T}_{t,\cdot}^*$. If in each source domains we have infinity number of training samples, then statistically Theorem 4.2 recovers the true $\mathcal{T}_{t,\cdot}^*$. In this case, we only need to observe $T = \text{poly}(p, K, d, M)$ source domains.

## 4.3 Main Theorem

We are now ready to show our main result.

**Theorem 4.4.** *Let assumption 4.1 holds and $w^* = \{w_1^* \dots, w_T^*\}$ being $(\nu, \epsilon)$-diverse for representation $\phi^*$. With probability at least $1 - 3\delta$, the following holds:*

$$\frac{1}{D}\sum_{t=1}^{D}\mathop{E}_{(x,y)\sim\mathcal{D}_t}[\ell(\hat{w}_t\circ\hat{\phi}(x),y) - \ell(w_t^*\circ\phi^*(x),y)] \le \frac{LD_{\mathcal{X}}W}{T}\sum_{t\in[T]}\sum_{j\in[p]}\left|\hat{w}_{t,j} - \tilde{\mathcal{T}}_{t,j}\right|$$

$$+ LD_{\mathcal{X}}Wp\sqrt{\frac{KdM^2\log(8ed) + \log(p/\delta)}{T}} + \tilde{\mathcal{O}}\left(\frac{C(\mathcal{W})}{n} + \frac{C(\Phi)}{nT}\right)^{1/4}.$$

The first two terms correspond to the cost of tensor completion and the last term corresponds to predicting with inaccurate $\hat{\phi}$. The term $\sum_{t\in[T]}\sum_{j\in[p]}\left|\hat{w}_{t,j} - \tilde{\mathcal{T}}_{t,j}\right|$ can be exactly computed on the training dataset. Recall $\hat{w}_t$ is the learned by eq. (2) and $\hat{\mathcal{T}}_{t,\cdot}$ is the estimated linear classifier after tensor completion. As $n$ increases, this difference becomes smaller as $\hat{w}_t$ becoming close to $w_t^*$.

## 5 EXPERIMENTS

We empirically verify that our proposed model leads to better generalization performance than baseline model with vanilla representation learning on three datasets: a variant of MNIST dataset, the ground terrain in outdoor scene (GTOS) dataset (Xue et al., 2017), and a fiber sensing dataset. On all datasets, we use LeNet trained on all available training data as the baseline. According to Li et al. (2017; 2019), this is a simple yet strong baseline that outperforms most domain generalization methods. Our model is trained in an end-to-end fashion. Instead of finding ERMs in eq. (2) and perform tensor completion, we directly represent $\hat{w}_t = \sum_{k=1}^{K} \odot_{m=1}^{M} \hat{\alpha}_{k,t_m}$, and output

$$\hat{\alpha}_{k,t_m}, \hat{\phi} = \underset{\substack{\phi \in \Phi, \\ \alpha_{k,t_m}}}{\arg \min} \frac{1}{nT} \sum_{t=1}^{T} \sum_{i=1}^{n} \ell \left( (\sum_{k=1}^{K} \overset{M}{\underset{m=1}{\odot}} \alpha_{k,t_m} \circ \phi)(x_{t,i}), y_{t,i} \right). \tag{5}$$

In this way, we can fully exploit the computational convenience of auto-differentiation rather than dealing with the algorithmic difficulty of tensor completion. All models are trained using the cross entropy loss. Since our theorem is a uniform convergence bound by nature, the sample complexity will also apply to the end-to-end trained model[1]. To prevent overfitting, we stop training of all models on the two-way MNIST dataset as soon as the last $50$ iterations have average loss less than $0.05$, and the training of all models on GTOS and the four-way fiber sensing dataset is stopped once the last $100$ iterations have average loss less than $0.05$. Throughout the experiments, the Adam optimizer with default learning rate $10^{-3}$ is used except when explicitly stated otherwise. The sensitivity of the regularization parameter is investigated on MNIST data and we set it to $0.05$ for all rest of the experiments. For MNIST dataset, the source and target domain data are augmented from MNIST training and test data respectively. For the fiber sensing dataset, the source and target domain data are collected in separate experimental rounds. The GTOS experiment results are postponed to the appendix. Across all results tables, mean accuracy is outside the parenthesis, standard deviation is inside the parenthesis.

### 5.1 TWO-WAY MNIST

A variant of MNIST is created by rotation and translation operations. We rotate all MNIST digits by $[-30, -15, 0, 15, 30]$ degrees, and translate all MNIST by $[(-3, 0), (0, -3), (0, 0), (0, 3), (3, 0)]$ pixels, leading to $5 \times 5$ domains.

For our proposed method, we use a simplified low-rank structure on the last two layers of LeNet. Specifically, LeNet has the structure of conv1-pool1-conv2-pool2-fc1-relu-fc2-relu-fc3-sigmoid. We impose the low-rank structure on both fc2 and fc3.

We create 11 linear classifiers for each layer, denote as $s_1, \ldots, s_5, v_1, \ldots, v_5, u$. For task $(i, j) \in 5 \times 5$, we use $s_i + v_j + u$ for prediction. This formulation is just a special case of the general formulation in eq. (1). Indeed, let $\alpha_1 = [s_1, s_2, s_3, s_4, s_5, \mathbf{1}, \mathbf{1}, \mathbf{1}, \mathbf{1}, \mathbf{1}]$, $\alpha_2 = [\mathbf{1}, \mathbf{1}, \mathbf{1}, \mathbf{1}, \mathbf{1}, v_1, v_2, v_3, v_4, v_5]$, and $\alpha_3 = [u, u, u, u, u, \mathbf{1}, \mathbf{1}, \mathbf{1}, \mathbf{1}, \mathbf{1}]$, where each $w_i, v_i, u, \mathbf{1} \in \mathbb{R}^p$ and $\alpha_k \in \mathbb{R}^{10} \times \mathbb{R}^p$.

Then for any task at $t = (i, j) \in 5 \times 5$, its classifier $w_t$ this can be formally written as $w_t = \sum_{k=1}^{3} \alpha_{k,i} \odot \alpha_{k,5+j}$, which is the same form as eq. (1). This is done for both fc2 and fc3, and each layer has its own distinct set of weights.

Similar idea has been previously proposed for domain generalization Yang & Hospedales (2017); Li et al. (2017). These previous works do not assume a tensor structure on the tasks. Instead, they put a low-rank tensor structure on the classifiers themselves. This fundamentally distinguishes our settings from previous ones. As a result, during test phase they have to use the common classifier $u$ for different target domains, but we can incorporate more prior knowledge by using $s_i + v_j + u$.

During training time, we collect training data from $(i, i)$ entries for all $i \in [5]$, and leave data in any other domains for test only. This is one of the designs requires the minimal number of source domains, and we can still successfully train all unknown classifiers $s$, $v$ and $u$. Due to

---

[1]Some theorems require $\hat{w}_t, \hat{\phi}$ being the ERM of eq. (2). In those cases, we can simply modify $\hat{w}_t$ to be the corresponding entries of the tensor $\widehat{\mathcal{T}}$ formed by $\hat{\alpha}_{k,t_m}$ and substitute eq. (5) for eq. (2). This is justified since we assume the learning problem is realizable.

Table 1: Test accuracy with 5 observed domains on the diagonal. In each cell, from the 1st to 5th row: baseline, our domain-agnostic and domain-specific models (both with the special low-rank formulation) the general ZSDA model (Yang & Hospedales, 2014), and the Fish algorithm (Shi et al., 2021) for domain generalization.

|  | (-3, 0) | (0, -3) | (0,0) | (0,3) | (3,0) |
|---|---|---|---|---|---|
| -30 |  | 0.958(0.007) | 0.927(0.007) | 0.735(0.017) | 0.585(0.016) |
|  |  | 0.950(0.007) | 0.932(0.010) | 0.769(0.024) | 0.659(0.037) |
|  |  | 0.965(0.004) | **0.943(0.009)** | **0.775(0.024)** | 0.646(0.038) |
|  |  | 0.967(0.002) | 0.936(0.006) | 0.769(0.022) | **0.671(0.021)** |
|  |  | **0.968(0.003)** | 0.932(0.004) | 0.744(0.018) | 0.637(0.012) |
| -15 | 0.975(0.003) |  | 0.974(0.002) | 0.908(0.005) | 0.797(0.009) |
|  | **0.978(0.003)** |  | 0.973(0.004) | 0.907(0.010) | 0.846(0.018) |
|  | 0.977(0.004) |  | 0.975(0.003) | 0.911(0.010) | 0.839(0.015) |
|  | 0.976(0.005) |  | 0.974(0.004) | **0.913(0.012)** | **0.852(0.012)** |
|  | **0.978(0.002)** |  | **0.977(0.001)** | 0.912(0.007) | 0.845(0.007) |
| 0 | 0.925(0.012) | 0.969(0.004) |  | 0.973(0.002) | 0.935(0.007) |
|  | **0.951(0.008)** | 0.966(0.005) |  | 0.971(0.004) | 0.947(0.007) |
|  | 0.950(0.009) | **0.971(0.005)** |  | 0.976(0.002) | **0.952(0.005)** |
|  | 0.945(0.011) | **0.971(0.004)** |  | **0.977(0.001)** | **0.952(0.004)** |
|  | 0.928(0.006) | **0.971(0.002)** |  | **0.977(0.002)** | 0.950(0.003) |
| 15 | 0.739(0.038) | 0.861(0.023) | 0.974(0.003) |  | 0.975(0.003) |
|  | **0.810(0.029)** | 0.863(0.013) | 0.971(0.007) |  | 0.975(0.002) |
|  | 0.801(0.027) | 0.866(0.013) | **0.978(0.003)** |  | 0.977(0.002) |
|  | 0.804(0.019) | **0.882(0.017)** | **0.978(0.001)** |  | 0.978(0.002) |
|  | 0.756(0.021) | 0.871(0.009) | 0.976(0.003) |  | **0.980(0.002)** |
| 30 | 0.494(0.039) | 0.649(0.027) | 0.919(0.010) | 0.955(0.004) |  |
|  | **0.576(0.048)** | 0.664(0.018) | 0.917(0.021) | 0.930(0.012) |  |
|  | 0.573(0.045) | 0.681(0.024) | **0.942(0.008)** | 0.956(0.006) |  |
|  | 0.568(0.022) | **0.715(0.018)** | 0.938(0.008) | 0.947(0.016) |  |
|  | 0.518(0.024) | 0.672(0.015) | 0.928(0.007) | **0.958(0.004)** |  |

the excess amount of learnable parameters, it is easy to overfit on our method. To reduce model complexity, we regularize all learnable classifiers to be close to their mean. That is, on each low-rank layer, let $\mu = \frac{1}{11}\left(\sum_i v_i + \sum_j s_j + u\right)$, we add the following regularizer to the loss function

$$\Omega_\lambda(s, v, u) = \frac{\lambda}{11}\left(\sum_i \|v_i - \mu\|^2 + \sum_j \|s_j - \mu\|^2 + \|u - \mu\|^2\right).$$

We run the experiments 10 times and report the mean performances and standard deviations in Table 1. Since this method uses the domain description information $(i, j)$ during testing, we refer to it as the *domain-specific* model. In addition, we also report the performance of using just the common classifier $u$ in fc2 and fc3. This model uses no domain information during testing, and we call it the *domain-agnostic* model. The baseline model is LeNet trained with data pooled from all source domains together. Notice that each $s_i + v_j + \nu$ corresponds to a unique task descriptor $q_{i,j}$ that serves as the coefficients for the combination of the linear layers, so for the ZSDA model, we further trained another descriptor network that outputs $\mathrm{ReLU}(Wq)$ as the new coefficients. This is done for both the last two layers. Our method almost uniformly dominates the baseline in every domain, sometimes by a large margin, and almost matches the performance of the ZSDA model, with less parameters. The domain-agnostic model achieves comparable performances to our proposed method. This shows that with the low-rank tensor structure, domain-agnostic models can also accommodate the domain shifts on this dataset. On most test domains, especially the ones that are "further away" from the training domains, our proposed algorithm consistently outperforms Fish (Shi et al., 2021), one of the state of the art domain generalization algorithms. Such results are expected as the Fish algorithm does not consider multi-way structure into consideration.

Another interesting observation is that the performances of all models are better near the diagonal, and getting worse away from the diagonal. This may provide insight into how we should design experiments under a fixed budget of data collection. A conjecture is that the performances on unseen target domains relate to the Manhattan distance of these domains to the seen source domains. Further discussion is deferred to the appendix.

## 5.2 Four-way Fiber Sensing Dataset

The distributed optic fiber sensing technique turns underground cable of tens of kilometers into a linear array of sensors, which could be used for traffic monitoring in smart city applications. In this paper, we aim to build a classifier for automatic vehicle counting and run-off-road event detection.

The objective is to classify whether the sensed vibration signal is purely ambient noise, or it contains vibration caused by a vehicle either driving normally or crossing the rumbling stripes alongside the

road. The sensing data takes the form of a 2D spatio-temporal array that can be viewed as an image - each pixel represents the vibration intensity at a particular time and location along the cable. Vehicles driving on the road, running-off-road, and ambient noise all create different patterns, which makes convolutional neural networks a natural choice for a 3-class classification model.

In experimental design, we consider several influencing factors representing the practical challenges faced after field deployment. These factors include weather-ground conditions (sunny-dry, rainy-wet), shoulder type (grass, concrete), sensing distance (0km, 10km, 15km), and vehicle type (van, sedan, truck), which produce combined effects on the data domain. In order to evaluate our approach, we spent efforts on collecting data with all possible combinations of the levels in the aforementioned factors in a lab testbed, resulting to a multiway data indexed by a $2 \times 3 \times 3 \times 2$ tensor. Under each condition, we conduct $20 \sim 25$ rounds for the run-off-road events and normal driving. The ambient noise data is collected when no vehicles are near the cable. The classes are balanced in both training and test data.

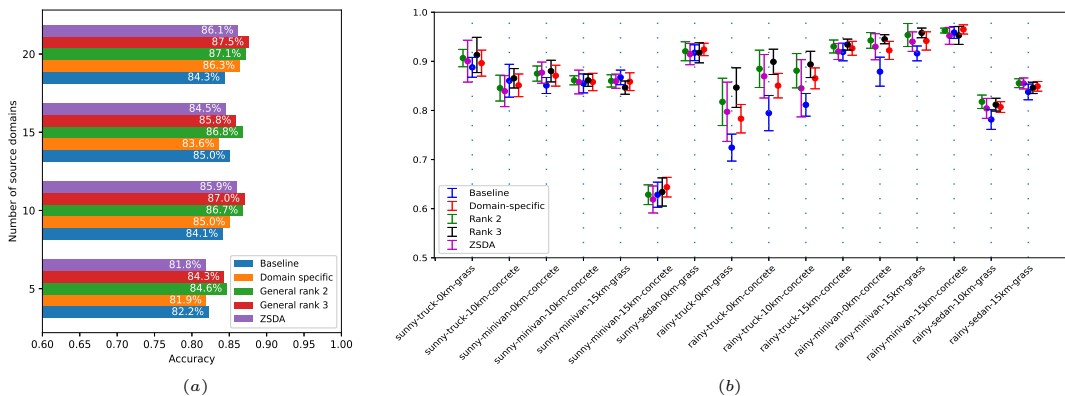

Figure 1: **Left:** Average test accuracy vs. number of source domains. **Right:** Test accuracy, trained on 20 random observed domains.

We evaluate model performance against number of source domains ($5, 10, 15$, or $20$, each with $10$ simulation runs) and results are shown in Figure 1(a). The performance increases with number of source domains as expected. In particular, Figure 1(b) summarizes the test performance for models trained with 10 source domains and tested on the rest. In each run, the observed source domains are randomly sampled. Among the 10 runs, "rainy-sedan-15km-grass" happens to be always selected as the source domain, thus the test result is not available for this domain.

We add two additional low-rank models in the form of eq. (5) with $K = 2$ and $3$ just in the last layer for comparison. Among the combination of factor levels, some scenarios are more challenging than others. For example, sedan is the lightest vehicle among the three, which generates much weaker vibration than a truck. Also the signal-to-noise ratio decays with increasing sensing distance. Results show that in most of the cases, our ZSDA models achieve improved performance over the baseline, and the margins over baseline are particularly large in several challenging scenarios.

## 6 CONCLUSION

In this work, we propose a particular domain adaptation framework where $T$ out of $D = d^M$ total domains are observed during training. All $D$ domains are parameterized by a common latent representation and their domain-specific linear functionals, which form a $d^{\times M}$-dimensional low-rank tensor. This multilinear structure allows us to achieve an average excess risk of order $\tilde{\mathcal{O}}\left(\left(\frac{TC(\mathcal{W})+C(\Phi)}{nT}\right)^{1/4} + p\left(\frac{KdM^2}{T}\right)^{1/2}\right)$. In addition to domain adaptation, our setting also sheds light on more efficient experiment designs and data augmentations. Algorithms developed under our framework are empirically verified on both benchmark and real-world datasets.

ACKNOWLEDGEMENT

SSD acknowledges support from NEC. SH would like to thank Yuheng Chen, Ming-Fang Huang, and Yangmin Ding from NEC Labs America for their help with the fiber sensing data collection.

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

## A  PROOFS IN SECTION 4

**Theorem 4.2.** *Let $\hat{w}, \hat{\phi}$ be the ERM in eq. (2). Let assumption 4.1 holds. With probability at least $1 - \delta$, we have*

$$\frac{1}{T}\sum_{t\in[T]}\|\hat{w}_t - w_t^*\|_2 = \mathcal{O}\left(\sqrt{\frac{2}{\lambda}}\left(L\left[\frac{WD_\mathcal{X}}{(nT)^2} + \log(nT)\cdot\left[W\cdot\mathfrak{G}_{nT}(\Phi) + \overline{\mathfrak{G}}_n(\mathcal{W})\right]\right] + 8B\sqrt{\frac{\log(2/\delta)}{nT}}\right)^{1/2}\right).$$

*Proof.* By a standard decomposition, we have with probability at least $1 - 2\delta$

$$\frac{1}{T}\sum_{t\in[T]}\left(L(\hat{w}_t\circ\hat{\phi}) - L(w_t^*\circ\phi^*)\right)$$

$$=\frac{1}{T}\sum_{t\in[T]}\left(\mathbb{E}_{\mathcal{D}_t}[\ell(\hat{w}_t\circ\hat{\phi})] - \mathbb{E}_{\mathcal{D}_t}[(w_t^*\circ\phi^*)]\right)$$

$$\leq\frac{1}{T}\sum_{t\in[T]}\mathbb{E}_{\mathcal{D}_t}[\ell(\hat{w}_t\circ\hat{\phi})] - \frac{1}{nT}\sum_{t\in[T]}\sum_{i\in[n]}\ell((\hat{w}_t\circ\hat{\phi})(x_{t,i}),y_i)$$

$$+\frac{1}{nT}\sum_{t\in[T]}\sum_{i\in[n]}\ell((\hat{w}_t\circ\hat{\phi})(x_{t,i}),y_i) - \frac{1}{nT}\sum_{t\in[T]}\sum_{i\in[n]}\ell((w_t^*\circ\phi^*)(x_{t,i}),y_i)$$

$$+\frac{1}{nT}\sum_{t\in[T]}\sum_{i\in[n]}\ell((w_t^*\circ\phi^*)(x_{t,i}),y_i) - \mathbb{E}_{\mathcal{D}_t}[\ell(w_t^*\circ\phi^*)]$$

$$\leq 2\sup_{\substack{w_1,\ldots,w_T\in\mathcal{W},\\ \phi\in\Phi}}\left|\frac{1}{T}\sum_{t\in[T]}\mathbb{E}_{\mathcal{D}_t}[\ell(w_t\circ\phi)] - \frac{1}{nT}\sum_{t\in[T]}\sum_{i\in[n]}\ell((w_t\circ\phi)(x_{t,i}),y_i)\right|$$

$$\leq 4\mathfrak{R}_{nT}(\ell(\mathcal{W}^{\otimes T}\circ\Phi)) + 4B\sqrt{\frac{\log(1/\delta)}{nT}}$$

$$\leq 8L\mathfrak{R}_{nT}(\mathcal{W}^{\otimes T}\circ\Phi) + 8B\sqrt{\frac{\log(1/\delta)}{nT}}$$

Use the fact that $\widehat{\mathfrak{R}}_n(\mathcal{H}) \leq \sqrt{\pi/2}\widehat{\mathfrak{G}}_n(\mathcal{H})$ for any function class $\mathcal{H}$ and the decomposition of $\mathfrak{R}_{nT}(\mathcal{W}^{\otimes T}\circ\Phi)$ introduced in Tripuraneni et al. (2020), we conclude that

$$\frac{1}{T}\sum_{t=1}^T\left(L(\hat{w}_t\circ\hat{\phi}) - L(w_t^*\circ\phi^*)\right)$$

$$\leq \underbrace{4096L\left[\frac{WD_\mathcal{X}}{(nT)^2} + \log(nT)\cdot\left[W\cdot\mathfrak{G}_{nT}(\Phi) + \overline{\mathfrak{G}}_n(\mathcal{W})\right]\right] + 8B\sqrt{\frac{\log(2/\delta)}{nT}}}_{\text{①}}$$

Meanwhile, by our assumption of uniform absolute representation difference and $(\nu, \epsilon)$-task diversity:

$$\frac{1}{T}\sum_{t=1}^T\left(L(\hat{w}_t\circ\phi^*) - L(\hat{w}_t\circ\hat{\phi})\right) \leq \frac{1}{T}\sum_{t=1}^T\left|L(\hat{w}_t\circ\phi^*) - L(\hat{w}_t\circ\hat{\phi})\right| \leq d_\mathcal{W}(\hat{\phi};\phi^*) \leq \bar{d}_{\mathcal{W},w^*}(\hat{\phi},\phi^*)/\nu + \epsilon,$$

which gives

$$\frac{1}{T}\sum_{t=1}^T(L(\hat{w}_t\circ\phi^*) - L(w_t^*\circ\phi^*)) = \frac{1}{T}\sum_{t=1}^T\left(L(\hat{w}_t\circ\phi^*) - L(\hat{w}_t\circ\hat{\phi})\right) + \frac{1}{T}\sum_{t=1}^T\left(L(\hat{w}_t\circ\hat{\phi}) - L(w_t^*\circ\phi^*)\right)$$

$$\leq \text{①} + \text{①}/\nu + \epsilon,$$

where we use $d_{\mathcal{W}, w^*}(\hat{\phi}, \phi^*) \leq \text{①}$ from Theorem 4.1.

By strong convexity, we conclude for any $T \in [D]$, $\nu \leq 1$:

$$
\frac{1}{T} \sum_{t \in [T]} \|\hat{w}_t - w_t^*\|_2 \leq \sqrt{\frac{2}{\lambda}} \left( \frac{1}{T} \sum_{t \in [T]} \left( L(\hat{w}_t \circ \phi^*) - L(w_t^* \circ \phi^*) \right) \right)^{1/2}
$$

$$
\leq \sqrt{\frac{4}{\lambda \nu}} \left( 4096L \left[ \frac{W D_{\mathcal{X}}}{(nT)^2} + \log(nT) \cdot \left[ W \cdot \mathfrak{G}_{nT}(\Phi) + \overline{\mathfrak{G}}_n(\mathcal{W}) \right] \right] + 8B \sqrt{\frac{\log(2/\delta)}{nT}} + \epsilon \right)^{1/2}.
$$

$\square$

Now we proceed to show the results related to tensor completion. The main idea is to treat a tensor as a polynomial, and count its connect components. This number restricts the complexity of the tensor.

**Corollary A.1.** *The number of $\{\pm 1, 0\}$ sign configurations of $r$ polynomials of degree at most $d$, over $q$ variables, is at most $(8edr/q)^q$ for $r > q > 2$.*

**Lemma 4.1.** *Let $\mathcal{X}_K$ be the class of rank-$K$ tensor of shape $d^{\times M}$, its pseudo-dimension can be bounded by $\text{Pdim}(\mathcal{X}_K) \leq KdM^2 \log(8ed)$.*

*Proof.* Let $f_T(d, M, K) = \left| \{ \text{sign}(X - T) \in \{\pm 1, 0\}^{d^{\times M}} : X \in \mathcal{X}_K \} \right|$. It suffices to show that $f_T(d, M, K) \leq (8ed)^{KdM^2}$. A rank $K$ $d^{\times M}$ tensor can be decomposed as:

$$
X_t = \sum_{k=1}^{K} \prod_{m=1}^{M} U_{k, t_m}
$$

for $t \in [d]^{\times M}$. Then one can treat $X - T$ as $d^M$ polynomials of degree at most $M$ over the following entries:

$$
(X - T)_t = \sum_{k=1}^{K} \prod_{m=1}^{M} U_{k, t_m} - T_t.
$$

$T$ is a fixed arbitrary tensor, so there are in total $KdM$ variables. Applying Corollary A.1 yields the desired result. $\square$

**Theorem 4.3.** *With probability at least $1 - \delta$,*

$$
\frac{1}{D} \sum_{t \in [D]} \left\| \widehat{\mathcal{T}}_{t, \cdot} - \mathcal{T}_{t, \cdot}^* \right\| \leq \frac{1}{T} \sum_{t \in [T]} \sum_{j \in [p]} \left| \widehat{\mathcal{T}}_{t,j} - \tilde{\mathcal{T}}_{t,j} \right| + p \sqrt{\frac{KdM^2 \log(8ed) + \log(p/\delta)}{T}} + \tilde{\mathcal{O}}(n^{-1/4}).
$$
(4)

*Proof.* The following equation holds with probability at least $1 - \delta$, and follows directly from the uniform convergence bound using covering number and bounding covering number using pseudo-dimension. See Srebro & Shraibman (2005) for detail.

$$
\frac{1}{D} \sum_{t \in [D]} \left| \widehat{\mathcal{T}}_{t,j} - \mathcal{T}_{t,j}^* \right| \leq \frac{1}{T} \sum_{t \in [T]} \left| \widehat{\mathcal{T}}_{t,j} - \mathcal{T}_{t,j}^* \right| + \sqrt{\frac{KdM^2 \log(8ed) - \log \delta}{T}}.
$$
(6)

Then by triangle inequality and equivalence of norms in finite-dimensional spaces

$$\frac{1}{D} \sum_{t \in [D]} \left\| \widehat{\mathcal{T}}_{t,\cdot} - \mathcal{T}_{t,\cdot}^* \right\| \leq \frac{1}{D} \sum_{t \in [D]} \sum_{j \in [p]} \left| \widehat{\mathcal{T}}_{t,j} - \mathcal{T}_{t,j}^* \right|$$

$$\leq \frac{1}{T} \sum_{t \in [T]} \sum_{t \in [p]} \left| \widehat{\mathcal{T}}_{t,j} - \mathcal{T}_{t,j}^* \right| + p \sqrt{\frac{KdM^2 \log(8ed) + \log(p/\delta)}{T}}$$

$$\leq \frac{1}{T} \sum_{t \in [T]} \sum_{t \in [p]} \left| \widehat{\mathcal{T}}_{t,j} - \tilde{\mathcal{T}}_{t,j} \right| + p \sqrt{\frac{KdM^2 \log(8ed) + \log(p/\delta)}{T}} + \frac{1}{T} \sum_{t \in [T]} \sum_{t \in [p]} \left| \tilde{\mathcal{T}}_{t,j} - \mathcal{T}_{t,j}^* \right|$$

$$\leq \frac{1}{T} \sum_{t \in [T]} \sum_{j \in [p]} \left| \widehat{\mathcal{T}}_{t,j} - \tilde{\mathcal{T}}_{t,j} \right| + p \sqrt{\frac{KdM^2 \log(8ed) + \log(p/\delta)}{T}} + \mathcal{O}(p^{1/2} n^{-1/4}),$$

where the first step is from the fact that $\| \cdot \|_2 \leq \| \cdot \|_1$ and union bounding eq. (6) over $p$ events. The $\mathcal{O}(p^{1/2} n^{-1/4})$ term follows from Theorem 4.2 and equivalence between norms. Note that $p \ll n$ hence asymptotically our conclusion holds.

$\square$

**Theorem 4.4.** *Let assumption 4.1 holds and $w^* = \{w_1^* \ldots, w_T^*\}$ being $(\nu, \epsilon)$-diverse for representation $\phi^*$. With probability at least $1 - 3\delta$, the following holds:*

$$\frac{1}{D} \sum_{t=1}^{D} \mathop{E}_{(x,y) \sim \mathcal{D}_t} [\ell(\hat{w}_t \circ \hat{\phi}(x), y) - \ell(w_t^* \circ \phi^*(x), y)] \leq \frac{LD_{\mathcal{X}} W}{T} \sum_{t \in [T]} \sum_{j \in [p]} \left| \hat{w}_{t,j} - \tilde{\mathcal{T}}_{t,j} \right|$$

$$+ LD_{\mathcal{X}} W p \sqrt{\frac{KdM^2 \log(8ed) + \log(p/\delta)}{T}} + \tilde{\mathcal{O}} \left( \frac{C(\mathcal{W})}{n} + \frac{C(\Phi)}{nT} \right)^{1/4}.$$

*Proof.*

$$\frac{1}{D} \sum_{t=1}^{D} \mathop{E}_{(x,y) \sim \mathcal{D}_t} [\ell(\hat{w}_t \circ \hat{\phi}(x), y) - \ell(w_t^* \circ \phi^*(x), y)]$$

$$= \underbrace{\frac{1}{D} \sum_{t=1}^{D} \mathop{E}_{(x,y) \sim \mathcal{D}_t} [\ell(\hat{w}_t \circ \hat{\phi}(x), y) - \ell(\hat{w}_t \circ \phi^*(x), y)]}_{A} + \underbrace{\frac{1}{D} \sum_{t=1}^{D} \mathbb{E}_{\mathcal{D}_t} [\ell(\hat{w}_t \circ \phi^*(x), y) - \ell(w_t^* \circ \phi^*(x), y)]}_{B}$$

Let $\bar{d}_{\mathcal{W}, w^*}(\hat{\phi}, \phi^*)$ be as defined in definition 4.1. By our assumption of uniform absolute representation difference and $(\nu, \epsilon)$-diverse, we can upper bound $A$ by $\bar{d}_{\mathcal{W}, w^*}(\hat{\phi}, \phi^*)/\nu + \epsilon$. The second term can be bounded by lipschitzness,

$$B \leq \frac{1}{D} \sum_{t \in [D]} \mathbb{E}_{\mathcal{D}_t} [L | \hat{w}_t \circ \phi^*(x) - w_t^* \circ \phi^*(x)|]$$

$$\leq \frac{1}{D} \sum_{t \in [D]} \mathbb{E}_{\mathcal{D}_t} [L \|\hat{w}_t - w_t^*\| \|\phi^*(x)\|]$$

$$\leq \frac{LD_{\mathcal{X}}}{D} \sum_{t \in [D]} \|\hat{w}_t - w_t^*\|.$$

Plug in our approximation to $d_{\mathcal{W}, w^*}(\hat{\phi}, \phi^*)$ in Theorem 4.1, $\frac{1}{D} \sum_{t \in [D]} \|\hat{w}_t - w_t^*\|$ in Theorem 4.4, and union bound, we conclude the theorem. $\square$

Table 2: Mean test accuracy for our method (both domain-specific and domain-agnostic) and baseline. Same settings as in Table 1 but difference domains are observed.

| ROTATION \ TRANSLATION | (-3, 0) | (0, -3) | (0,0) | (0,3) | (3,0) |
|---|---|---|---|---|---|
| -30 | | 0.740(0.021)
0.767(0.024)
0.796(0.022) | 0.932(0.009)
0.940(0.008)
0.947(0.008) | 0.960(0.005)
0.958(0.007)
0.965(0.003) | 0.604(0.025)
0.654(0.044)
0.644(0.043) |
| -15 | 0.971(0.003)
0.977(0.003)
0.977(0.003) | 0.906(0.012)
0.915(0.007)
0.914(0.007) | 0.976(0.002)
0.978(0.002)
0.980(0.002) | | 0.829(0.016)
0.860(0.021)
0.866(0.019) |
| 0 | 0.919(0.011)
0.953(0.006)
0.950(0.007) | 0.973(0.004)
0.972(0.003)
0.975(0.003) | | 0.969(0.003)
0.961(0.007)
0.969(0.005) | 0.948(0.006)
0.952(0.006)
0.955(0.006) |
| 15 | 0.756(0.024)
0.844(0.020)
0.813(0.019) | | 0.966(0.005)
0.969(0.005)
0.967(0.008) | 0.861(0.016)
0.844(0.024)
0.853(0.025) | 0.978(0.002)
0.976(0.002)
0.977(0.003) |
| 30 | 0.554(0.021)
0.657(0.029)
0.656(0.028) | 0.958(0.006)
0.956(0.005)
0.966(0.005) | 0.903(0.021)
0.914(0.018)
0.940(0.011) | 0.656(0.025)
0.632(0.027)
0.691(0.030) | |

## B  MORE DISCUSSION TO SECTION 5

We mention in Section 5.1 that the test accuracy on unseen domain might relate to the Manhattan distance between the seen and unseen domains. Here is another experiment in the same setting as in section 5.1, but the chosen observed entries are [(0,0), (1,3), (2,2), (3,1), (4,4)].

In both Table 2 and Table 1, every factor level has been observed exactly once. If the factors are categorical nominal, then permuting rows 2 and row 4 in Table 2 leads to the same balanced design with Table 1, and the performance shall be similar. However, the factors (rotation, translation) considered here are not nominal, for example, there is a ordinal relation between rotation $-30$ and rotation $-15$. Hence, the above-mentioned permutation seems prohibitive. Consequently, it makes sense to talk about Manhattan distances between observed and unobserved entries.

In Table 1, cell (4,0) is 4 units away from its closest observed cell in Manhattan distance (in the following we omit to mention the metric is Manhattan distance), but in Table 2, (4,0) is 2 units away from the closest observed entry. One can see the accuracy in that domain gets much better. However, the closest distance to the observed entry is not the only factor here. For example, (0,1) in Table 1 outperforms (0,1) in Table 2 a lot. Therefore, the average distance to the observed entries may also be a contributing factor.

It is an open question how to select the best subset of tensor entries to be observed, such that the overall performance in the unseen target domains can be optimized. This question may relate to the area of factorial experiment design, where many factors are involved and some have confounded interactions. One subject is to design minimal sets of experiments such that the effects of all factors can be studied. Our theory deals particularly with the case when the factor combinations are chosen uniformly at random, without assuming any particular tensor structure. With more prior knowledge, more efficient sampling algorithms can be designed. For instance, McKay et al. (2000) discussed how to use Latin hypercube sampling to achieve a smaller sample complexity. There are also some works that connect the dots between error coding theory and factorial experiment design Ben-Gal & Levitin (2001). There could be information theoretic understanding to the relation between our setting of experiment design and Manhattan distance as well. This can be an interesting research direction and beyond the scope of our paper.

**Fish Algorithm**  The Fish algorithm was proposed in Shi et al. (2021) as a domain generalization method, which achieves state-of-the-art results on the WILDS benchmarks. Its intuition is that the model should be regularized (implicitly) during training such that the gradient steps on different domains are invariant, thus leading to an invariant feature representation across different domain. Comparing to the Fish algorithm, our proposed algorithm consistently outperforms Fish on most test domains, especially the ones that are "further away" from the training domains. Such results

are expected as the Fish algorithm does not consider multi-way structure into consideration. See the results in table 1.

Fish is used for the LeNet architecture. Its inner step is trained using an Adam optimizer with step size $10^{-2}$ and the meta step uses a learning rate of $0.5$. The number of meta steps is set to be $5$. We use the same hyperparameters in the GTOS experiment as well. During training, we found out that the Fish algorithm does not converge well if the batch size is too large. Hence we pick batch size $20$. This is of the same magnitude as the batch size in Shi et al. (2021), but our proposed model and the ERM models exhibit better generalization when trained with larger batch size $200$.

**Hyperparameter Sensitivity**    Model selection in domain adaptation is tricky in general, since no data from the target domains is seen. One benefit of using the special low-rank formulation is that it has fewer number of tuning parameters than general low-rank formulations. In addition to tuning $\lambda$ in the regularizer $\Omega_\lambda$, the form in eq. (5) also requires to choose rank $K$. We evaluate the sensitivity of $\lambda$ on both the source and target domains. Observing data from $(i, i)$ domains on the diagonal, we train on $5000$ training data using $\lambda \in [0.005, 0.01, 0.03, 0.05, 0.1, 0.5, 1]$, and test on $1000$ data from both $(i, i)$ source domains for $i \in [5]$ and $(i, j)$ task domains for $i \neq j \in [5]$. Results show that the test performances on both source domains and target domains are insensitive to $\lambda$. The mean performances and standard deviations are reported in Table 3.

Table 3: Hyperparameter sensitivity.

| $\lambda$ | *Source Domain* | *Target Domain* |
|---|---|---|
| 0.005 | 0.963 (0.003) | 0.822 (0.005) |
| 0.01 | 0.964 (0.003) | 0.821 (0.010) |
| 0.03 | 0.962 (0.003) | 0.824 (0.006) |
| 0.05 | 0.963 (0.003) | 0.818 (0.007) |
| 0.1 | 0.959 (0.003) | 0.823 (0.008) |
| 0.5 | 0.961 (0.005) | 0.810 (0.010) |
| 1 | 0.962 (0.004) | 0.811 (0.001) |

## C    THREE-WAY GROUND TERRAIN IN OUTDOOR SCENES DATASET

To train a model invariant to domain shifts, one would either need to collect massive training data from many different domains such that the model is able to generalize (implicitly) to unseen cases, or build a mechanism allowing (explicitly) adaptation to new scenarios utilizing limited domain information. Our work belongs to the latter.

The proposed multi-way domain setup represents a combination of the environmental or experimental factors, which naturally arises in many real life applications and the levels of factors are pre-known. This setup is general enough for many tasks, especially those involving data collection experiments that are costly and limited by physical constraints. To make the applicability of our method more compelling, we present results on the GTOS (Ground Terrain in Outdoor Scenes) dataset (Xue et al., 2017), which a large-scale real-world dataset that naturally fits into our framework.

The task of outdoor ground material recognition is strongly influenced by the weather and lighting conditions, as well as the surface viewpoints. The GTOS dataset consists of over $30,000$ images covering $40$ classes (cement, asphalt, brick, etc.). The multi-way domain structure include: $4$ different weather conditions (cloudy dry, cloudy wet, sunny morning and sunny afternoon), $3$ illumination conditions with different exposure times, and $19$ viewpoints from differential angular imaging. As a result, each domain is indexed on a multidimensional array of size $4 \times 3 \times 19$. Experimental results in table 4 show that our method outperforms both the ERM and the Fish algorithm.

For our model, we use the same architecture as described in the beginning of section 5.1: a simplified low-rank structure on the last two layers of LeNet. Specifically, LeNet has the structure of conv1-pool1-conv2-pool2-fc1-relu-fc2-relu-fc3-sigmoid. We impose the low-rank structure on both fc2 and fc3. We create 27 linear classifiers for each layer, denote as $s_i, u_j, v_k, h$, for $i \in [3], j \in [4], k \in$

[19]. For task $(i, j, k) \in [3] \times [4] \times [19]$, we use $s_i + u_j + v_k + h$ for prediction. The ERM model and Fish model both use LeNet.

Both our model and the ERM model are trained using Adam optimizers with learning rate $10^{-3}$ with batch size 200. The Fish model was trained with batch size 20, its inner step is trained with an Adam optimizer with learning rate $10^{-2}$ and the meta step has learning rate 0.5. We choose 5 for the meta steps $N$ (this is the parameter that determines how many training domains the model sees at every iteration. The original paper tested $N = 5, 10, 20$ and found the performance insensitive to $N$).

During training, we found that Fish tends to not converging with large batch size, hence we reduce the batch size from 200 to 20. For Fish to converge, we also need to increase the learning rate. One conjecture of this phenomenon is because the number of observed training domains is quite large, hence it is hard to match their gradients. As Shi et al. (2021) pointed out, Fish usually does not outperform ERM when the number of domains is enormous. This is consistent with our observation.

Table 4: Experiments on GTOS

| % of observed entries | ERM | Ours | Fish |
|---|---|---|---|
| 0.1 | 0.602(0.059) | **0.663(0.091)** | 0.565(0.022) |
| 0.2 | 0.729(0.031) | **0.794(0.058)** | 0.653(0.083) |
| 0.3 | 0.792(0.013) | **0.905(0.021)** | 0.731(0.032) |
| 0.4 | 0.800(0.021) | **0.913(0.013)** | 0.759(0.040) |
| 0.5 | 0.884(0.037) | **0.946(0.008)** | 0.851(0.025) |

