# OpenReview forum: "Provable Adaptation across Multiway Domains via Representation Learning"
_ICLR.cc/2022/Conference — ICLR 2022 Poster_

### Official Review · Reviewer_mkWr · 2021-10-26

**Correctness:** 4
**Technical Novelty And Significance:** 3
**Empirical Novelty And Significance:** 3
**Recommendation:** 6
**Confidence:** 4

**Main Review:**

The paper is very well written, with very few typos, and technically sound. Nonetheless, I think that some aspects should be further clarified. Specifically:

(-) Despite the gains in computational efficiency, what are the disadvantages, if any, of replacing the optimization problem in eqs. (2)-(3) by the surrogate in eq. (5)? Is it just a constraint on the space of admissible solutions?

(-) In the experiments with the MNIST dataset, the authors create linear classifiers on two network layers, namely fc2 and fc3. How are the predictions of the two layers combined to produce the final decision? Moreover, what would be the effect of building the classifiers only on top of the last layer (fc3)?

(-) In the same experiment (MNIST), what is the value of $K$? More importantly, how was it chosen and what is the effect of varying it?

My only concerns about the paper are:

i) The applicability of the method seems a bit limited. I don't see a very large class of practical applications where the domains can be represented by a vector of attributes and hence fit into the proposed framework.

ii) The experimental results are not totally convincing. The method should be compared with at least one SOTA algorithm for domain generalization, e.g. Matsuura and Harada (2020).


Refs:

Matsuura, Toshihiko, and Tatsuya Harada. "Domain generalization using a mixture of multiple latent domains." Proceedings of the AAAI Conference on Artificial Intelligence. Vol. 34. No. 07. 2020.

**Summary Of The Paper:**

The paper addresses the problem of domain generalization in the setting where each domain is indexed on a multidimensional array. It assumes that only a limited number of domains is observed during training and presents a technique for learning a classifier with provable good generalization performance on the unseen domains. The method consists of learning a common latent feature representation for all domains and domain-specific linear functionals, which are obtained by imposing a low-rank constraint on the tensor built by aggregating the functionals of all domains. The authors present a thorough theoretical analysis of the algorithm, including an upper bound for the excess risk on the target domain.

**Summary Of The Review:**

Although I have some doubts about the applicability of the method to practical applications, I don't see this as a reason for rejection. Moreover, the paper definitely has significant novelty and is technically sound. I would like to see an improved experimental section to increase my score.

---

> ### Author Response · Authors · 2021-11-21
> **Thanks for your review**
>
> We thank the reviewer for your valuable comments.
> > what are the disadvantages, if any, of replacing the optimization problem in eqs. (2)-(3) by the surrogate in eq. (5)? Is it just a constraint on the space of admissible solutions?
>
> There should be no practical disadvantages of replacing eq. (2)-(3) by eq. (5). In terms of the uniform convergence theorem, these two are exactly the same. The spaces of learnable functions in these two equations are the same: as in eq(1), the set of $\alpha$'s exactly formulates the space of rank-$K$ tensor. However, these two different loss functions may have different implicit biases. It is an open problem on how to quantitatively characterize different implicit biases, and we think it is an interesting future direction.
>
> >  In the experiments with the MNIST dataset, the authors create linear classifiers on two network layers, namely fc2 and fc3. How are the predictions of the two layers combined to produce the final decision? Moreover, what would be the effect of building the classifiers only on top of the last layer (fc3)?
>
> The performance of this mentioned model is dataset-dependent. For example, on the two-way MNIST dataset, the test accuracy (averaged over all test domains) of the last two layers combined vs the last layer combined is roughly the same; on the fiber-sensing data, the test accuracy (averaged over all test domains) of the last layer combined degrades about 1-2%. Hence we pick the last two layers combined model for consistency.
>
> >  In the same experiment (MNIST), what is the value of  K? More importantly, how was it chosen and what is the effect of varying it?
>
> In the MNIST dataset, we use a heuristic low-rank structure described at the beginning of section 5.1 (2nd paragraph). It can be interpreted as a special rank-3 structure. For tuning hyperparameters, in our setting, given $T$ training domains, one has the freedom to divide it into $A$ training domains and $B$ evaluation domains where $A+B=T$. As long as $A$ is big enough, the performance will not degrade too much. One can then pick the best set of hyperparameters on the $B$ evaluation domains. Although we also observe that in our experiment, test accuracy is not highly sensitive to hyperparameters (see figure 1 for varying $K$ and table 3 in the appendix for varying the regularizer $\lambda$).
>
> > The applicability of the method seems a bit limited. I don't see a very large class of practical applications where the domains can be represented by a vector of attributes and hence fit into the proposed framework.
>
> We agree applicability of our method is more restricted than the general domain generalization model. However, we argue that there are many important applications that do permit a tensor structure. *We have discussed the applicability of our method in the second and the third paragraph of Introduction.* Furthermore, in our revision, *we also tested our method on a new dataset, Ground Terrain Outdoor Scene (GTOS) [Xue et al. 2017]* to further demonstrate the applicability of our method. See Section C in Appendix or the **Summary of Revision** thread for more details.
>
> > The experimental results are not totally convincing. The method should be compared with at least one SOTA algorithm for domain generalization,
>
> Thank you for your suggestion! We have tested Fish (Shi et al. 2021), a SOTA method on WILDS dataset, on Two-Way MNIST and GTOS datasets. See the **Summary of Revisions** thread and discussions in Section 5.1 and Section C in the revised version.

---

> > ### Comment · Reviewer_mkWr · 2021-11-24
> > **Update after the authors' response**
> >
> > The authors have answered my questions and addressed my concerns adequately. I find the comparison with the Fish algorithm and the additional experiments on the GTOS dataset particularly valuable. These experiments show that the proposed algorithm outperforms Fish significantly when the domain shift and the number of domains are large. These observations reinforce my opinion that the paper deserves acceptance.

---

### Official Review · Reviewer_Qy5s · 2021-11-02

**Correctness:** 4
**Technical Novelty And Significance:** 2
**Empirical Novelty And Significance:** 2
**Recommendation:** 3
**Confidence:** 5

**Main Review:**

Strengths: While the theoretical results are interesting, the underlying assumptions are quite restrictive and will not hold for many computer vision tasks.
Weaknesses; Results on MNIST and the fiber sensing dataset are vastly inadequate. There are more interesting datasets for evaluating domain generalization methods, such as WILDS. The WILDS leaderboard lists a few SOTA results.

**Summary Of The Paper:**

This paper proposes a specific domain adaptation framework where a subset of all possible domains is are observed during training. Given sufficient training samples for the observed source domains, a common latent representation as well as a domain specific linear classifier is learned. Theoretical conditions under which the learned representations will be effective for unseen domains are provided. Experiments using two datasets are provided.

**Summary Of The Review:**

Unless I see results on more challenging datasets such as WILDS, I will remain unconvinced of the usefulness of this work. It is not just the number of source domains T that matter.

---

> ### Author Response · Authors · 2021-11-21
> **Thanks for your review**
>
> We thank you for your valuable comments.
>
> >WILDS
>
> WILDS dataset is a good benchmark for domain generalization. However, the WILDS dataset **does not permit the multi-way structure** and thus our algorithm cannot be applied here.
> Nevertheless, we have added new experiments using the more challenging GTOS dataset which fits our setting. See the **Summary of Revisions** thread and Appendix C in the revised version for details.
> Furthermore, we also added another baseline FISH by Shi et al. [1], which is a SOTA method for domain generalization, and tested on two-way MNIST and GTOS. See the **Summary of Revisions** thread and section 5.1 in the revised version for details.
> [1] Gradient matching for domain generalization.

---

### Official Review · Reviewer_b2ge · 2021-11-03

**Correctness:** 3
**Technical Novelty And Significance:** 4
**Empirical Novelty And Significance:** 2
**Recommendation:** 8
**Confidence:** 2

**Main Review:**

Positive aspects:
- The error bounds presented appear correct, to the best of my knowledge.
- The authors provide some empirical evidence of the benefits of their approach, on two different datasets. The results appear good, demonstrating the interest of their approach.
- The paper is well written, clear.

Concerns:
- Concerning the empirical evaluation, I would note that conducting additional experiments on datasets more complex than MNIST-derived ones (and the sensing dataset) would be beneficial to show that the method works well in a more realistic setting. Is there another dataset that the authors could scale to, such as adapting the CelebA dataset to a similar task as the MNIST experiments?
- Could the authors provide more details on how the different hyper-parameters were determined?

**Summary Of The Paper:**

The paper considers the problem of zero-shot domain adaptation. The contributions are two-fold. The authors propose a novel domain adaptation technique and provide bounds on the prediction error, which they provide proofs for. In addition, they provide empirical validation of their proposed approach, on two sets of datasets: one derived from MNIST, the second corresponding to sensors.

**Summary Of The Review:**

I have not found flaws in the proofs or reasoning of the paper, and find the theoretical contributions particularly interesting. While I feel the experimental part is less impressive, I feel that this paper deserves acceptance. As noted in my confidence score, I am not overconfident about this, and would recommend weighing my opinion accordingly.

---

> ### Author Response · Authors · 2021-11-21
> **Thanks for your review**
>
> We thank the reviewer for your valuable comments.
>
> >More experiments
>
> Thanks for your suggestion. We have added an experiment on the GTOS dataset. See the **Summary of Revisions** thread and Appendix C in the revised version  for details.
>
>
> >Hyperparameter tuning
>
>  in our setting, given $T$ training domains, one has the freedom to divide it into $A$ training domains and $B$ evaluation domains where $A+B=T$. As long as $A$ is big enough, the performance will not degrade too much. One can then pick the best set of hyperparameters on the $B$ evaluation domains. Although we also observe that in our experiment, test accuracy is not highly sensitive to hyperparameters  (see figure 1 for varying $K$ and table 3 in the appendix for varying the regularizer $\lambda$).

---

> > ### Comment · Reviewer_b2ge · 2021-12-06
> > **Response to reviewer comments**
> >
> > I would like to thank the authors for responding to my comments. One of my initial concerns was the lack of evaluation on datasets more complex than the MNIST-derived one. The authors have now provided such results, and comments regarding my question on hyperparameter tuning. As a result, I have raised my score to an accept (8).

---

### Official Review · Reviewer_5mcf · 2021-11-09

**Correctness:** 4
**Technical Novelty And Significance:** 2
**Empirical Novelty And Significance:** 2
**Recommendation:** 6
**Confidence:** 3

**Main Review:**

Strengths:

- In my opinion, it is a good contribution for zero shot domain adaptation analysis, it can have good potential future work on the theoretical side.
- The authors support their analysis with small experiments and by showing the stability of their results. They obtain a bound on the polynomial scale with exponential input dimensions so it is satisfactory.

Weaknesses:

- The contributions are a little limited. The method is very similar with Tripuraneni et al. (2020) if I am not missing anything, but with a condition that not using any test tasks in the training. Also the authors define uniform absolute difference in order to keep distribution same with the true distribution.
- The total number of source and target domains need to be known beforehand, so it might not be directly used for a new target domain.
- The paper is not the first to do finite sample analysis in zero-shot domain, see [1]. You could rephrase your claim.
- The paper was well written(there are a couple of typos especially in the equations - such as in theorem 4.2, 3rd line should be $=$, not $\leq$.), however it made me go back and forth a lot(needs a bit organization). Its relation with previous work could be stated better, the analysis and method parts could be separated. Some of the definitions in section 4 can be carried to preliminary since they are borrowed from the previous work.

Questions:

- How does the correlations between the labels/domains affect your bounds? How do you quantify the similarity between the domains other than through classifier parameters?
- Did you test your algorithm on larger real data? You could try lower p and with not-very-different source and target domains.
- The method adds sum of linear functionals to a neural network with a couple of weight norm constraints, so I would expect results other than 30 degree rotated MNIST(although there are convexity and linearity assumptions in the theoretical part).
- In table 3, what did you choose as the value of K? How did you choose your experimental details, such as why 11 classes?

[1] Blitzer, John, Dean P. Foster, and Sham M. Kakade. "Zero-shot domain adaptation: A multi-view approach." Tech. Rep. TTI-TR-2009-1 (2009).

**Summary Of The Paper:**

Short summary:

The authors propose a zero-shot domain adaptation method with the assumption of each domain having the same set of labels. They obtain finite sample complexity bounds that contain the number of samples(n) per class in the training, the number of seen(training) and unseen(test) tasks under regularity assumptions on the loss and mappings. They provide accuracy results for MNIST and MNIST with rotated digits.

Method:

First, a common linear mapping is used to obtain a common representation for tasks. Then each domain-specific classifer(linear) combines at each index as a rank K tensor. A tensor completion is applied at each index that would give classifier parameters for unseen data as well, that allows them to predict without any data from the target domain. They assume same number of samples and same input dimensions for each domain (the latter can directly be extended to different dimensions). For the experiments, they use LeNet and directly predicted decomposed tensors rather than training and applying tensor completion.

Contributions:

The authors extend Tripuraneni et al. (2020) method and proofs from transfer learning to zero-shot case. They show a uniform convergence bound that has polynomial complexity with respect to the # tasks, # dimensions and rank. They derive excess risk on the order of n^-1/4.


**Summary Of The Review:**

In general, I am a little bit concerned about the significancy of the paper's contributions. The method and theory part shows a lot of similarity with Tripuraneni et al. (2020). However, bringing it and the analysis to zero shot domain adaptation is valuable in my opinion.

---

> ### Author Response · Authors · 2021-11-21
> **Thanks for your review.**
>
> We thank the reviewer for your valuable comments.
>
> >"Zero-shot domain adaptation: A multi-view approach.":
>
> Thanks for mentioning this paper. In our revised version, we have deleted our claim “we obtain the first finite-sample guarantee for ZSDA” and we have cited this paper in Section 2.
>
> > How does the correlations between the labels/domains affect your bounds? How do you quantify the similarity between the domains other than through classifier parameters?
>
> The correlation among domains is implicitly captured by the low-rank structure. Intuitively, the lower the rank, the stronger the correlation. The correlation among the labels is also implicitly captured by the multi-way structure, since we assume the true data-generating function has a low-rank structure.
>
> > Did you test your algorithm on larger real data?
>
> Thanks for your suggestion. We have added an experiment on the GTOS dataset. See the **Summary of Revisions** thread and Appendix C in the revised version for details.
>
> > The method adds the sum of linear functionals to a neural network with a couple of weight norm constraints, so I would expect results other than 30 degrees rotated MNIST(although there are convexity and linearity assumptions in the theoretical part).
>
> Thanks for your suggestion! We will consider adding experiments on this in the final version.
>
> > In table 3, what did you choose as the value of K?
>
> The regularizer in table 3 was based on the formulation mentioned at the beginning of Section 5.1 (2nd paragraph), so there is no $K$ to tune. The choice of 11 classifiers is a heuristic that one may want one classifier for each domain and an extra common one, then combine them linearly. Such a heuristic has been proposed before in the multi-task learning literature [1].
>
> [1]: Regularized Multi-Task Learning
>
> Writing: Thanks for your suggestions. We will further polish our paper in the final version.

---

### Author Response · Authors · 2021-11-21
**Summary of Revisions**

We greatly appreciate the valuable comments of all reviewers. All major revisions are marked in blue in the pdf. Two common concerns raised by many of the reviewers are 1) applicability of our setting on a large dataset, and 2) comparisons to SOTA domain generalization methods. We try to address both of them here.

1. We conduct additional experiments on the GTOS dataset. This dataset is a real-world dataset consisting of over $30,000$ images covering $40$ classes (cement, asphalt, brick, etc.). The multi-way domain structure includes $4$ different weather conditions (cloudy dry, cloudy wet, sunny morning, and sunny afternoon), $3$ illumination conditions with different exposure times, and $19$ viewpoints from differential angular imaging. These conditions naturally form a (3,4,19) multi-way task tensor. The goal is to correctly identify the surface material from different weather/illumination/angular conditions. Our model shows superior performance compared to other baselines. See the detail in the revision appendix C.

2. We choose the Fish algorithm from the WILDS leaderboard. This is a DG algorithm that achieves SOTA performances on the WILDS benchmark. We compare our method to Fish on both our 2-way MNIST (result in table 1, revision) and the GTOS dataset (appendix C, revision), and our method consistently outperforms the Fish algorithm, especially the ones that are “further away'' from the training domains.
One thing worth noticing is that on 2-way MNIST, Fish and our's have comparable test accuracy on task domains that are close to the training domains. However, as the test domains get further away from the training domains, Fish's accuracy drops drastically. This is strong evidence that a normal DG algorithm that ignores domain index information is not a good fit in our setting.

---

### Decision · Program_Chairs · 2022-01-20

**Decision:**

Accept (Poster)

**Comment:**

Thanks for your submission to ICLR.

This paper explores zero-shot adaptation from a theoretical perspective.  Three of the four reviewers are quite positive about the paper, particularly after the discussion phase.  One reviewer was more negative, citing a lack of compelling experiments and some possibly restrictive assumptions.

The authors responded to these concerns, as well as the concerns of the other reviewers.  One of the more positive reviewers increased their score from 6 to 8.  I did not hear from the negative reviewer, but my feeling is that I tend to agree with the authors that the focus of the paper is more on the theoretical side.  Moreover, the authors did add some additional results to the main paper, so I am of the opinion that the paper should indeed be accepted to the conference.

Even though this is paper is on the theoretical side, please do include as strong a set of empirical results as possible in the final version.  Also keep in mind the other suggestions from the reviewers when preparing the final manuscript.